# A Dual-Band High-Gain Subwavelength Cavity Antenna with Artificial Magnetic Conductor Metamaterial Microstructures

**DOI:** 10.3390/mi13010058

**Published:** 2021-12-30

**Authors:** Guang Lu, Fabao Yan, Kaiyuan Zhang, Yunpeng Zhao, Lei Zhang, Ziqian Shang, Chao Diao, Xiachen Zhou

**Affiliations:** 1School of Space Science and Physics, Shandong University at Weihai, Weihai 264209, China; 17862701682@163.com (K.Z.); 15588318820@163.com (Y.Z.); 201920773@mail.sdu.edu.cn (L.Z.); 201920775@mail.sdu.edu.cn (Z.S.); physicsdc@163.com (C.D.); xcsduwh@163.com (X.Z.); 2Laboratory for Electromagnetic Detection (LEAD), Institute of Space Sciences, Shandong University at Weihai, Weihai 264209, China; 3School of Mechanical, Electrical & Information Engineering, Shandong University at Weihai, Weihai 264209, China

**Keywords:** metamaterial, sub-wavelength cavity antenna, artificial magnetic conductor, dual-band, high gain

## Abstract

This paper presents dual-band high-gain subwavelength cavity antennas with artificial magnetic conductor (AMC) metamaterial microstructures. We developed an AMC metamaterial plate that can be equivalent to mu-negative metamaterials (MNMs) at two frequencies using periodic microstructure unit cells. A cavity antenna was constructed using the dual-band AMC metamaterial plate as the covering layer and utilizing a feed patch antenna with slot loading as the radiation source. The antenna was fabricated with a printed circuit board (PCB) process and measured in an anechoic chamber. The |S_11_| of the antenna was −26.8 dB and −23.2 dB at 3.75 GHz and 5.66 GHz, respectively, and the realized gain was 15.2 dBi and 18.8 dBi at two resonant frequencies. The thickness of the cavity, a sub-wavelength thickness cavity, was 15 mm, less than one fifth of the long resonant wavelength and less than one third of the short resonant wavelength. This new antenna has the advantages of low profile, light weight, dual-frequency capability, high gain, and easy processing.

## 1. Introduction

With the development of wireless communication technology, antennas for wireless communication require properties such as miniaturization, light weight, multifrequency capability, high gain, and other requirements. This creates new challenges for antenna theory and design. Metamaterials are a new class of artificial composites exhibiting exceptional properties that are not readily observed in nature [1,2,3,4,5,6,7,8]. Metamaterials consist of periodically or randomly distributed artificial structures that have a size and spacing much smaller than the wavelengths of incoming electromagnetic radiation. Their discovery provides a new theory and technical method for controlling electromagnetic waves and a new idea for the design of high-performance antennas [9,10,11,12,13].

Metamaterials include double-negative metamaterials (DNMs) and single-negative metamaterials (SNMs). SNMs include epsilon-negative metamaterials (ENMs) with negative permittivity and mu-negative metamaterials (MNMs) with negative permeability [1,14,15]. The reflection phase of perfect ENMs is π, which is called out-of-phase reflection. Metals are natural ENMs, and the reflection phase of smooth metal surfaces is nearly π. [16,17,18] Therefore, the minimum thickness of the cavity antenna composed of a smooth metal surface is one half of the resonant wavelength [19,20]. The reflection phase of perfect MNMs is zero, which is called in-phase reflection [21,22,23]. Perfect MNMs do not exist in nature; recent studies have shown that some artificial microstructures, known as artificial magnetic conductors (AMC), can achieve in-phase reflection [24,25,26,27,28,29,30,31]. By using the characteristic of in-phase reflection, a sub-wavelength resonator can be realized to break through the limit of the half-wavelength resonance scale, realizing the miniaturization of the microwave system [24,25]. Furthermore, the sub-wavelength cavity constructed by AMC metamaterials can be used to design a highly directional antenna with a low profile [26,27,28,29,30,31].

New miniaturized cavity antennas with AMC metamaterials have attracted a great deal of attention. However, AMC metamaterials are based on the local resonance of the electromagnetic field, which leads to AMC metamaterial antennas usually being single-frequency narrow-band antennas [26,27,28,29,30,31], which limits the further development and application of this type of antenna. In this paper, we propose an AMC metamaterial plate based on a multilayer structure which can be equivalent to perfect MNMs at two frequencies. Using a dual-band AMC metamaterial plate as the covering layer and a dual-band slot-loaded rectangular patch antenna as the feed antenna, a dual-band high-gain antenna can be realized. The cavity thickness of the antenna is subwavelength, and the antenna is a planar structure which is easy to integrate with other devices. The antenna adopts mature circuit board etching technology which is convenient to manufacture. The substrate integrated waveguide (SIW) technology is also a good method for achieving a dual-frequency antenna [32,33,34,35]. This method has the added benefit of miniaturization. In the existing research, this method has not considered such an antenna’s need for high gain.

The remainder of this paper is organized as follows. In Section 2, we introduce the proposed AMC metamaterials and their experimental preparation. In Section 3, we describe the construction of the cavity antenna using AMC metamaterials and the simulated radiation performance of the antenna, followed by preparation and subsequent performance measurement in a microwave anechoic chamber. Finally, our conclusions are provided in Section 4.

## 2. Materials and Methods

An F-P interferometer consists of two reflective surfaces that form a resonant cavity. Ray optics analysis is widely used to describe such systems [19,20]. The structure shown in Figure 1a is a typical F-P cavity model. A cavity is formed by two reflective layers separated by some distance. To achieve coherent enhancement of the reflected waves in the cavity, the following conditions must be met:
Δ*Φ* = *Φ_A_* + *Φ_B_* − (2π/*λ*) · 2*D* = 2Nπ, N = 0, 1, 2…   or *D* = (*Φ_A_* + *Φ_B_*) · (*λ*/4π) + N · (*λ*/2), N = 0, 1, 2…(1)
where *Φ_A_* and *Φ_B_* are the reflection phase shifts of layers *A* and *B*, respectively, *λ* is the free-space wavelength, and *D* is the length of the cavity. The typical reflection layer is an out-of-phase layer that causes a π reflection phase change on reflection (*Φ_A_* = *Φ_B_* = π, *Φ_A_* + *Φ_B_* = 2π). The thickness of the cavity should be *λ*/2 to satisfy condition (1). The typical resonant cavity can only achieve coherent enhancement at the resonant frequency and the multiple frequencies.

Phase control is a characteristic of metamaterials. Supposing that metamaterials are used to replace the reflective surfaces in a resonant cavity, in such a case the reflection phase in the cavity changes (*Φ**_A_* + *Φ_B_* ≠ 2π) and the thickness of the cavity can be less than *λ*/2 to satisfy condition (1). For example, if a reflective surface is replaced with an MNM metamaterial plate (*Φ_M_* = 0) and the sum of reflection phases is π (*Φ_M_* + *Φ_B_* = 0 + π = π), the thickness of cavity is reduced to *λ*/4. The subwavelength resonator is able to break through the limit of the half-wavelength resonance scale by using the characteristic of in-phase reflection. Metamaterials can also change the reflection phase at different frequencies to satisfy condition (1) and break typical cavity antennas’ single-frequency characteristics.

Metamaterials are composed of periodic subwavelength metallic/dielectric unit cells that resonantly couple to the electric and magnetic fields of the incident wave. Therefore, we first analyzed the electromagnetic properties of an AMC unit cell. We designed a dual-band AMC unit cell; Figure 2a,b shows the layered structure, and the structural parameters are shown in Table 1. The unit cell is composed of three layers. The first layer is a circular metal patch with diameter *a*. The second layer is a circular metal patch with diameter, *b*. There is a square dielectric block (ε = 2.6, tanδ = 0.01) between the first and second layer, with width *w*_1_, and thickness *d*_1_. The third layer is a square metal patch with a circular slot in the middle; the width of the square patch is *l*, and the diameter of the circular slot is *c*. There is a square dielectric block (ε = 2.6, tanδ = 0.01) between the second and third layer, with width *w*_2_ and thickness *d*_2_.

We calculated the electromagnetic characteristics of the AMC unit cell using CST Microwave Studio software. The magnetic and electric wall boundary conditions were enforced along the ±*x* and ±*y* directions to form a waveguide. Next, port 1 of the waveguide and its reference plane were added to the surface of the circular metal patch, and port 2 of the waveguide was positioned at the surface of the square metal patch with the circular slot. Hence, the excitation of port 1 was equivalent to a plane wave incident to the *z*-direction and had linear polarization along the *y*-direction. Figure 3a is the reflection phase diagram; the phase is 0° at 3.89 GHz and 5.66 GHz. The unit cell can be equivalent to MNMs near these two frequencies. Figure 3b shows the reflection diagram; the reflections are approximately 0.94 at two resonant frequencies. This indicates that the structure has the property of partial reflection. Figure 3c,d show the simulated electrical field distribution at 3.89 GHz and 5.66 GHz, respectively. The field has high resonance localization in the two dielectric layers between the first metal layer and the fifth metal layer at 3.89 GHz, while at 5.66 GHz it is mainly localized in the dielectric layer between the first metal layer and the third metal layer. The realization of the dual-band AMC is due to the resonant coupling of two cavities. The AMC metamaterial can be used to construct a dual-frequency subwavelength resonant cavity using the characteristics of a 0° reflection phase and partial reflection. The effective permeability (*µ_eff_*) and permittivity (*ε_eff_*) of the AMC unit cell can be calculated by the effective refractive index and impedance [36], as shown in Figure 4, where the black (blue) lines denote the real part of *ε_eff_* (*µ_eff_*) and the red (green) line denotes the imaginary part of *ε_eff_* (*µ_eff_*). It can be seen that the real part of *µ_eff_* is negative, while the real part of *ε_eff_* is positive at regions of gray background; the value of the real part of *µ_eff_* is relatively large, and negative near 3.89 GHz and 5.66 GHz.

Next, we calculated the effects of different structural parameters on the performance of AMCs. The reflection characteristic of an AMC is simulated by changing the size of a unit cell, as shown in Figure 5, and only one structural parameter in Table 1 is changed in each calculation. When the diameter *a* increases, the low resonance frequency shifts to a lower frequency and the reflection increases, while the high resonance frequency shifts to a lower frequency and the reflection decreases. When the diameter *b* increases, the low resonance frequency shifts to a lower frequency and the reflection is almost constant, while the high resonance frequency shifts to a lower frequency and the reflection increases. When the diameter *c* increases, the low resonance frequency shifts to a lower frequency and the reflection decreases, while the high resonance frequency is constant and the reflection decreases. When the thickness *d*_1_ increases, the low resonance frequency shifts to a lower frequency and the reflection increases, while the high resonance frequency shifts to a lower frequency and the reflection decreases. When the thickness *d*_2_ increases, the low resonance frequency shifts to a lower frequency and the reflection decreases, while the high resonance frequency and the reflection are almost constant. According to the above change rules, dual-band AMC unit cells can be designed at different frequencies.

## 3. Results and Discussion

### 3.1. Results for the Antenna

Figure 6 shows the geometry of the proposed cavity antenna operating at two frequencies. The overall volume of the antenna is 200 mm × 200 mm × 19 mm, and the thickness of the air cavity is 15 mm. The antenna includes three layers, as shown in Figure 6a,b. The top layer is a dual-band AMC reflector with a periodic array of AMC unit cells; the dimensions of the unit cell are shown in Table 1. The middle layer is the air-filled subwavelength cavity. The bottom layer is a 1.0 mm thick dielectric (ε = 2.6, tanδ = 0.01) substrate with a metallic patch feed covered on the upper face and a metallic ground plane as a reflector on the other face. A 22.9 mm × 19.6 mm rectangular microstrip feed patch antenna designed as a radiation source is embedded in the cavity at the center of the plane. Dual-frequency operation is achieved by modifying the natural modes through slot loading [37]; the parameters are shown in Table 2. Using commercial electromagnetic simulation software (CST Microwave studio) (CST Setting: 3 GHz–7 GHz, times domain solver, 30 cells per wavelength, 20 cells per max model box edge, 48,889,728 meshcells), we investigated the performance of the subwavelength cavity antenna with the planar dual-band AMC. The values for the antenna were calculated using a high-performance workstation (Intel(R) Xeon(R) Platinum 8270 CPU @ 2.70 GHz, 1.5 TB Memory (RAM), NVIDIA Quardro RTX 8000).

The simulated antenna reflection coefficient (|S_11_|) is shown in Figure 7a. There are two reflection dips less than −10 dB in the frequency range of 3 GHz to 6 GHz. |S_11_| are −28.7 dB and −22.6 dB at 3.73 GHz and 5.60 GHz, respectively. This shows that the antenna can radiate or receive electromagnetic waves at two frequencies. The antenna’s realized gain is shown in Figure 7b; the maximum realized gain of the metamaterial antenna is 15.5 dBi at 3.73 GHz in the frequency range of 3.5–3.9 GHz, while the realized gain of the feed antenna is only 7.6 dBi. In the frequency range of 5.4–5.8 GHz, the maximum gain of the metamaterial antenna is 19.2 dBi at 5.60 GHz, while the gain of the feed antenna is only 7.5 dBi. Figure 8 gives the gain pattern in the E- and H-planes. The antenna has high directivity performance at 3.73 GHz and 5.60 GHz. Beam width is 29.9° in the H-plane and 26.9° in the E-plane at 3.73 GHz. Beam width is 17.0° in the H-plane and 19.9° in the E-plane at 5.60 GHz. The thickness of the cavity is 15 mm, which is a sub-wavelength thickness cavity, less than one fifth of 80.4 mm (3.73 GHz) and less than one third of 53.6 mm (5.60 GHz). The simulation results show that a dual-band high-gain metamaterial subwavelength cavity antenna is realized. Low frequency and high frequency correspond to the same physical size and have different electrical sizes. The electrical size corresponding to high frequency is larger, so the antenna can have a greater gain at high frequencies than at low frequencies. The working frequency of the antenna is slightly different from the resonant frequency of AMC because the simulated AMC is an infinite period structure, while the AMC planar used in the antenna is a finite period structure. The effect of the period number on the properties of the metamaterials was studied in [28]. A higher period number led to a more ideal result, while a decrease in the number of cycles caused the resonance frequency to shift to high frequencies.

In order to analyze the influence of different structural parameters on antenna performance, we first analyzed the effect of layer misalignment on radiation performance. During the preparation process, the metamaterial was composed of two dielectric plates. Circular metal patches were attached to the top and bottom of a dielectric plate, and a square metal patch with a circular slot was attached to another plate. We analyzed the |S_11_| (Figure 9a) and realized gain (Figure 9b) of the antenna when the two dielectric plates were horizontally moved a distance m. When m ≤ 3 mm, the |S_11_| hardly changed. When m = 4 mm, the |S_11_| worsened at low frequencies, and hardly changed at high frequencies. When m ≤ 4, the realized gain of the antenna hardly changed. It can be seen that the structure was not sensitive to layer misalignment.

We then analyzed the effect of the thickness of the air cavity on radiation performance; the results can be seen in Figure 10. When the thickness of the air cavity *h* increased, the two dips of |S_11_| did not shift; rather, the value gradually increased, the maximum gain of the antenna shifted to low frequencies, and the value gradually decreased. When *h* decreased, the two dips of |S_11_| did not shift; instead they gradually increased in value, the maximum gain of the antenna shifted to high frequencies, and the value gradually decreased. It can be seen that the optimum antenna thickness is *h* = 15 mm. Increasing or decreasing the cavity thickness will weaken the radiation performance.

### 3.2. Experimental Results of the Antenna

The subwavelength cavity antenna was fabricated with a printed circuit board (PCB) process. Figure 11a,b shows the top view and bottom view of the AMC metamaterial plate. A 10 × 10 copper mesh array was placed on the top of a 1 mm thick dielectric (ε = 2.6, tanδ = 0.01) plate; a 10 × 10 copper circular patch array was placed on the bottom of a 2 mm thick dielectric (ε = 2.6, tanδ = 0.01) plate; and a copper circular patch array was placed on the middle of the two dielectric plates. Figure 11c shows the radiation source, where a single metallic patch with two rectangular slots is used as the radiation source on the top surface, and a 50 Ω input port on the opposite side. The overall volume of the antenna was 220 mm × 220 mm × 19 mm, and the thickness of the air cavity was 15 mm. The fabricated antenna was 20 mm wider than the designed antenna for convenient assembly. The radiation performance of the antenna was measured in an anechoic chamber using a far-field measurement system and KEYSIGHT PNA N5224B 10 MHz–43.5 GHz network analyzer. The test environment of the antenna is shown in Figure 12. The antenna was placed on a turntable that could rotate 360°. The emission source was a 2–18 GHz horn antenna.

The measured antenna reflection coefficient (|S_11_|) is shown in Figure 13a. There are two reflection dips less than −10 dB. |S_11_| is −26.8 dB and −23.2 dB at 3.75 GHz and 5.66 GHz, respectively. The realized gain is shown in Figure 13b. The maximum realized gain of the metamaterial antenna is 15.2 dBi at 3.75 GHz in the frequency range of 3.5–3.9 GHz, and the realized gain of the feed antenna is only 7.4 dBi. In the frequency range of 5.4–5.8 GHz, the maximum gain of the metamaterial antenna is 18.5 dBi at 5.60 GHz, and the gain of the feed antenna is only 7.3 dBi. Figure 14 shows the gain pattern in the E-plane and H-plane. The antenna has high directivity performance at 3.75 GHz and 5.66 GHz. Beam width is 27.7° in the H-plane and 31.5° in the E-plane at 3.75 GHz. Beam width is 18.8° in the H-plane and 22.5° in the E-plane at 5.66 GHz. The thickness of the cavity is 15 mm, which is a sub-wavelength thickness cavity, less than one fifth of 80.0 mm (3.75 GHz) and less than one third of 53.0 mm (5.66 GHz). The measured results are in good agreement with the simulated results. The small deviation is caused by manufacturing error and measurement error.

We measured the radiation performance of the antennas with different cavity thicknesses. Figure 15a shows |S_11_| and Figure 15b presents the realized gains. The measured results are in good agreement with the simulated results. When the thickness of the antenna cavity gradually increases or decreases, the |S_11_| gradually increases. When *h* increases, the maximum realized gain shifts to low frequencies and decrease gradually. When *h* decreases, the maximum realized gain shifts to low frequencies and decreases gradually. Finally, a comparison with other works in Table 3 shows that our studied dual-band antenna has the advantage of high gain.

## 4. Conclusions

In this study, a dual-band high-gain subwavelength cavity antenna was developed and studied. We obtained AMC metamaterials that can be equivalent to MNMs at two frequencies by using the principle of multiple resonance coupling. A subwavelength cavity antenna was constructed using this dual-band AMC metamaterial plate and a dual-band feed patch antenna. The antenna was fabricated and measured in a microwave anechoic chamber. The measured results are in good agreement with the simulated results. High gain is obtained at two frequencies. This new antenna has the advantages of low profile, light weight, dual-frequency capability, high gain and easy processing, and can be used in next-generation wireless communication systems.

## Figures and Tables

**Figure 1 micromachines-13-00058-f001:**
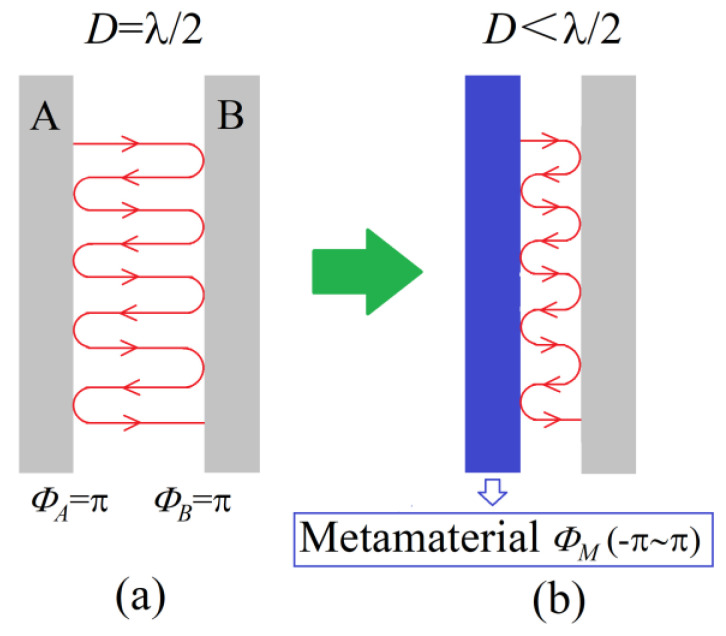
The schematic diagram of the structures of the cavity mode. (**a**) Traditional F-P cavity. (**b**) Metamaterial F-P cavity.

**Figure 2 micromachines-13-00058-f002:**
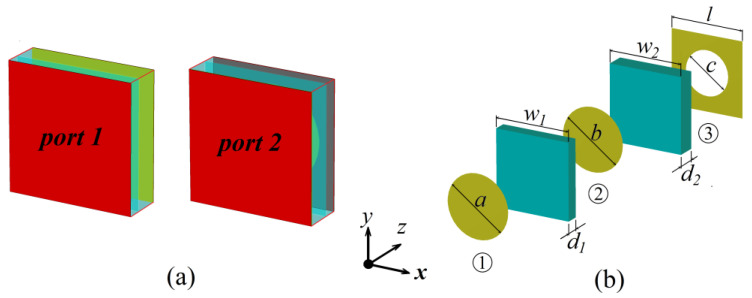
(**a**) Unit cell of a dual-band AMC where the electromagnetic waves are incident from port 1 to port 2. (**b**) Hierarchical diagram of the designed dual-band AMC.

**Figure 3 micromachines-13-00058-f003:**
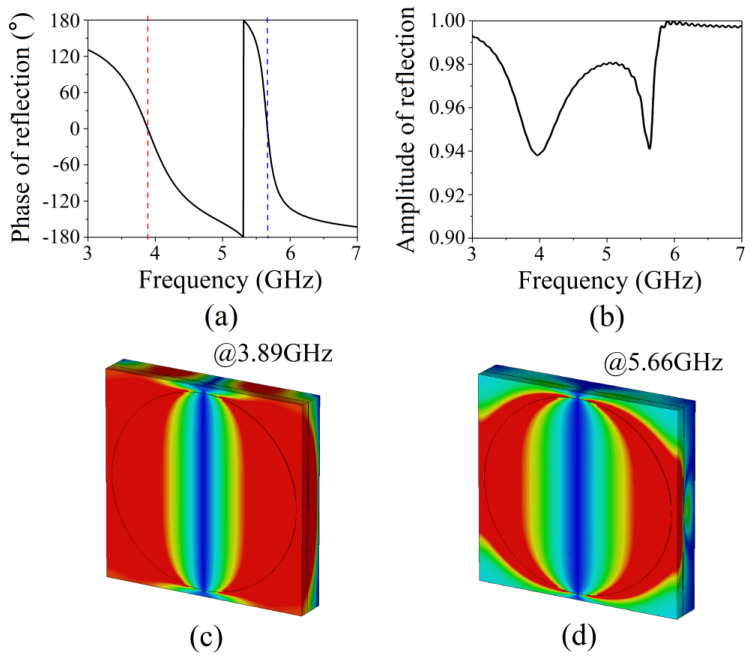
(**a**) Simulated reflection amplitude of a unit cell. (**b**) Simulated reflection phase of a unit cell. (**c**) Simulated electric field distribution at (**c**) 3.89 GHz and (**d**) 5.66 GHz.

**Figure 4 micromachines-13-00058-f004:**
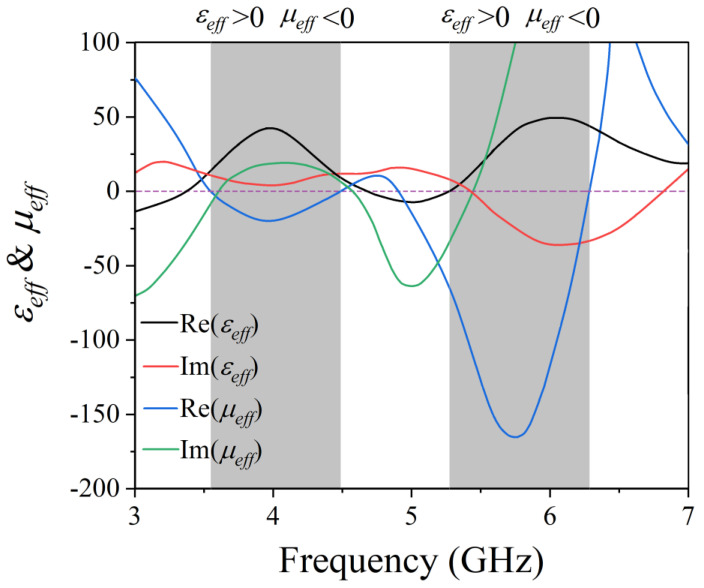
The effective permeability (*µ_eff_*) and permittivity (*ε_eff_*) of the AMC unit cell where the black (blue) lines denote the real part of *ε_eff_* (*µ_eff_*) and the red (green) line denotes the imaginary part of *ε_eff_* (*µ_eff_*).

**Figure 5 micromachines-13-00058-f005:**
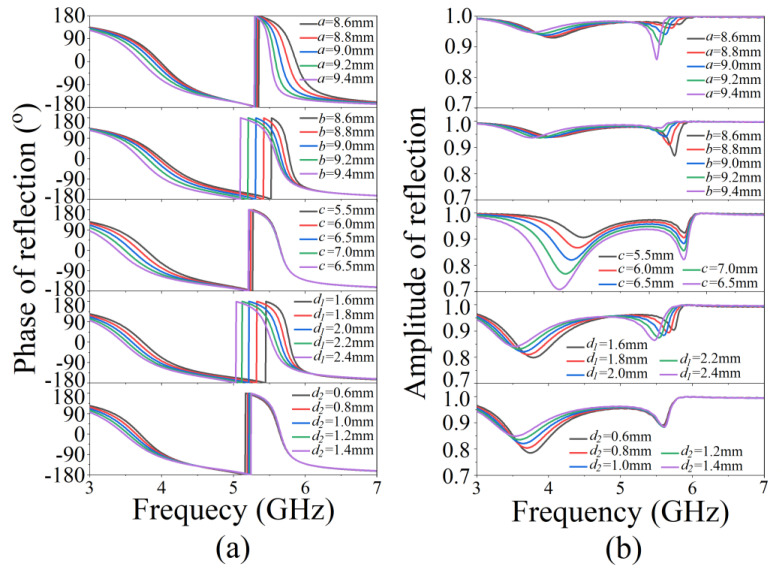
Simulated complex reflection coefficient with varying geometrical parameters: (**a**) phase and (**b**) amplitude for varying *a*, *b*, *c*, *d*_1_, and *d*_2_.

**Figure 6 micromachines-13-00058-f006:**
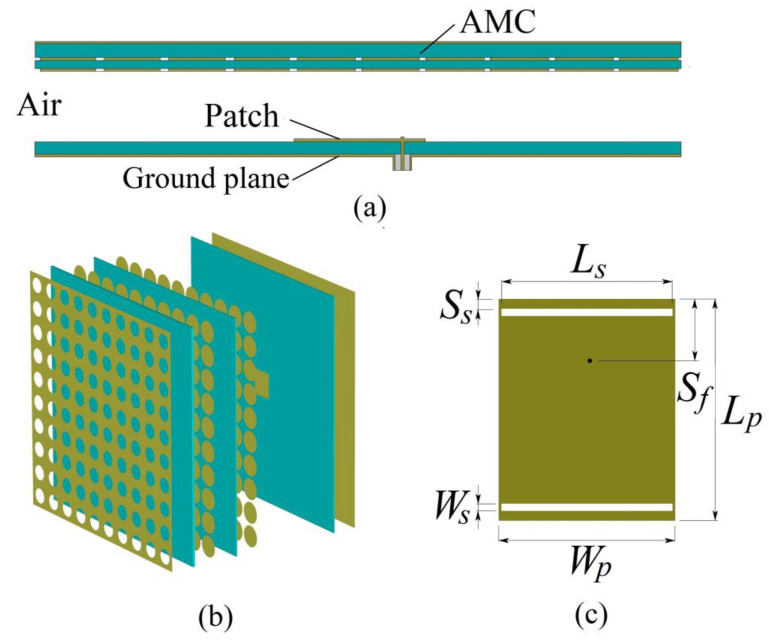
Geometry of the proposed antenna: (**a**) cross-sectional view, (**b**) 3–D view, and (**c**) feed patch antenna with slot loading.

**Figure 7 micromachines-13-00058-f007:**
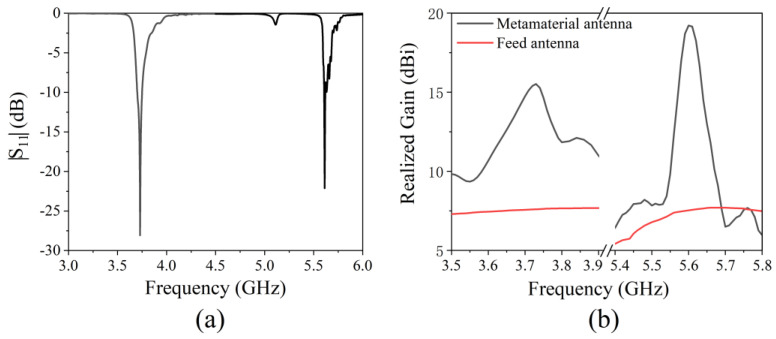
Simulated (**a**) return loss (|S_11_|) and (**b**) realized gain of the metamaterial antenna.

**Figure 8 micromachines-13-00058-f008:**
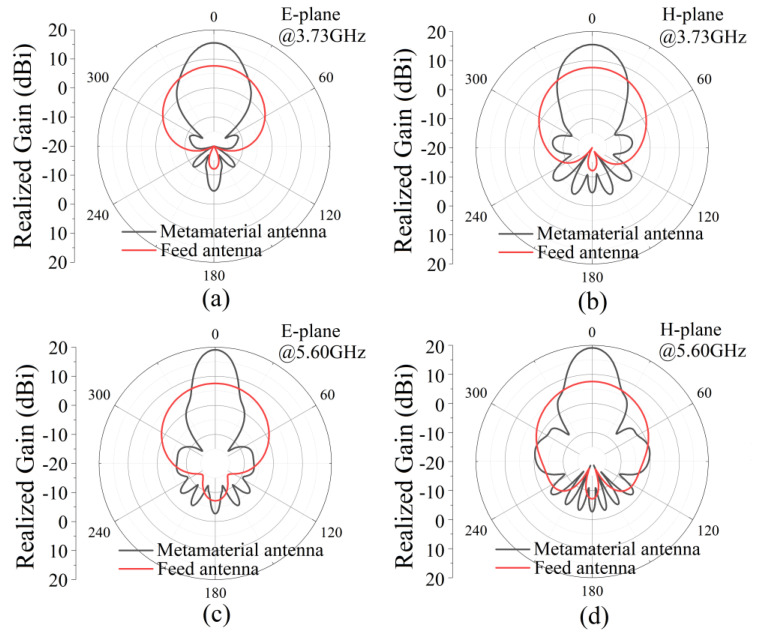
Simulated radiation patterns on (**a**,**c**) the E-plane and (**b**,**d**) the H-plane at 3.73 GHz and 5.60 GHz. The black lines are the results for the metamaterial antenna, and the red lines are the results for the feed patch antenna.

**Figure 9 micromachines-13-00058-f009:**
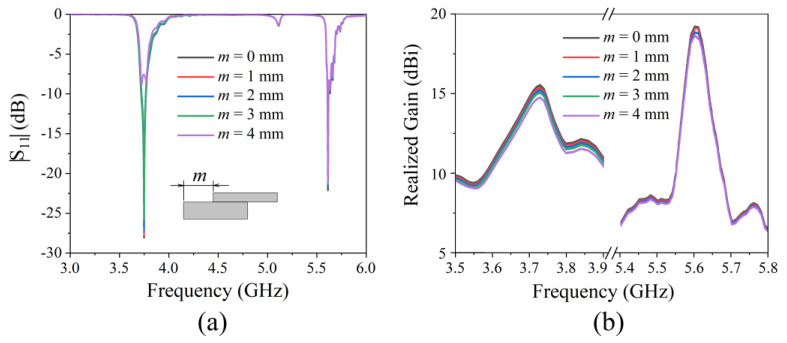
The effect of layer misalignment on radiation performance: (**a**) |S_11_|; (**b**) realized gain.

**Figure 10 micromachines-13-00058-f010:**
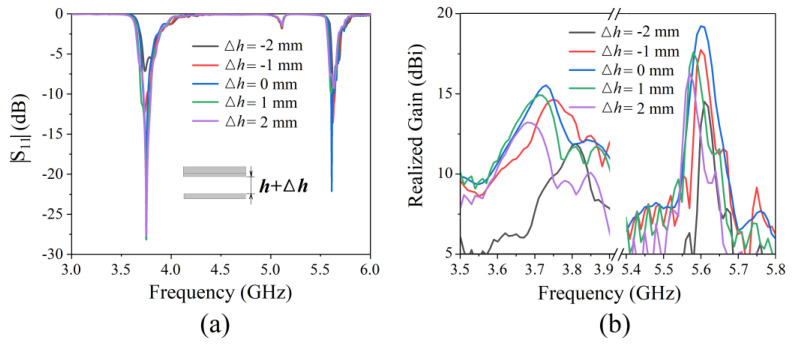
Simulated radiation performance for antennas with different cavity thicknesses: (**a**)|S_11_|; (**b**) realized gain.

**Figure 11 micromachines-13-00058-f011:**
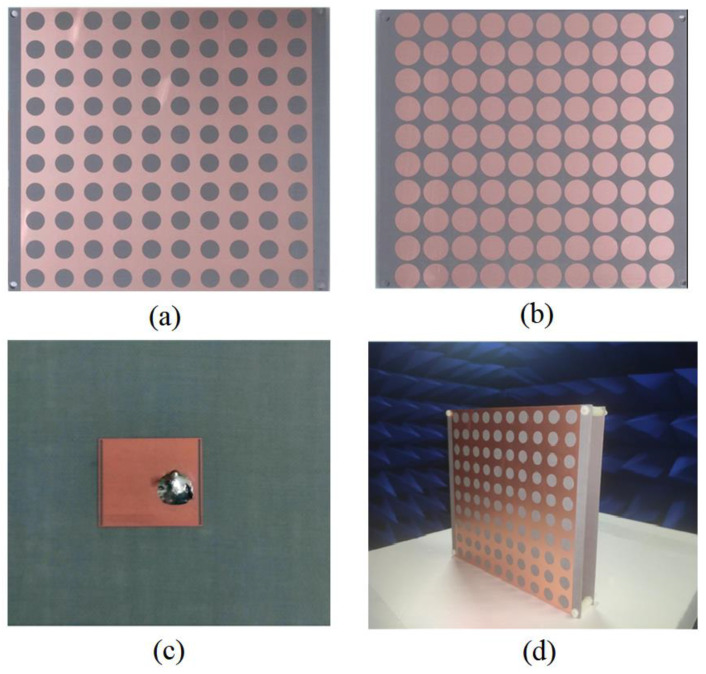
Photos of the metamaterial antenna: (**a**) the top view of the dual-band AMC plate; (**b**) the bottom view of the dual-band AMC plate; (**c**) feed patch antenna; (**d**) the metamaterial antenna placed in the microwave antrum.

**Figure 12 micromachines-13-00058-f012:**
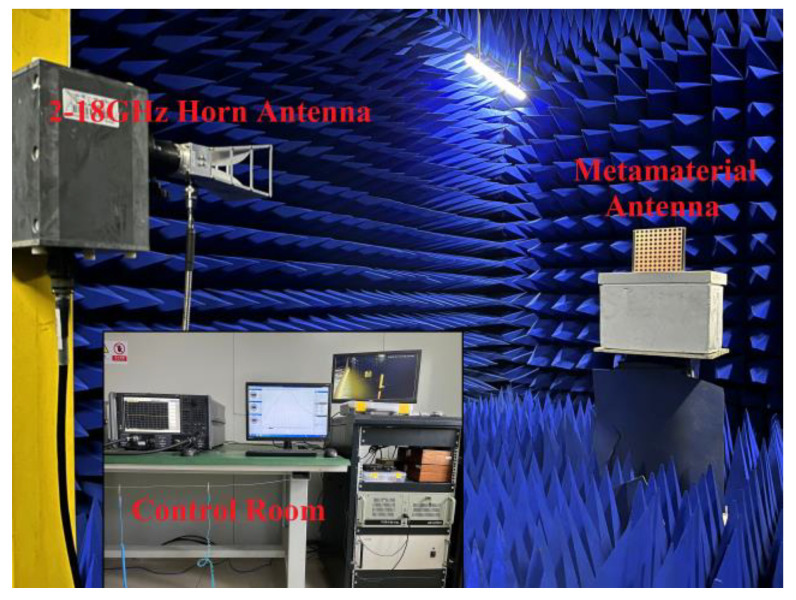
The test environment of the antenna.

**Figure 13 micromachines-13-00058-f013:**
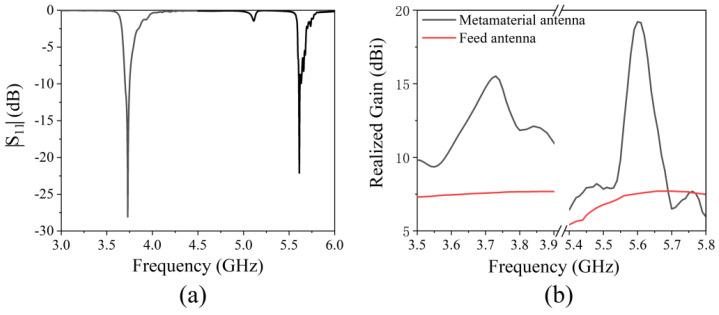
Measured (**a**) return loss (|S_11_|) and (**b**) realized gain of the metamaterial antenna.

**Figure 14 micromachines-13-00058-f014:**
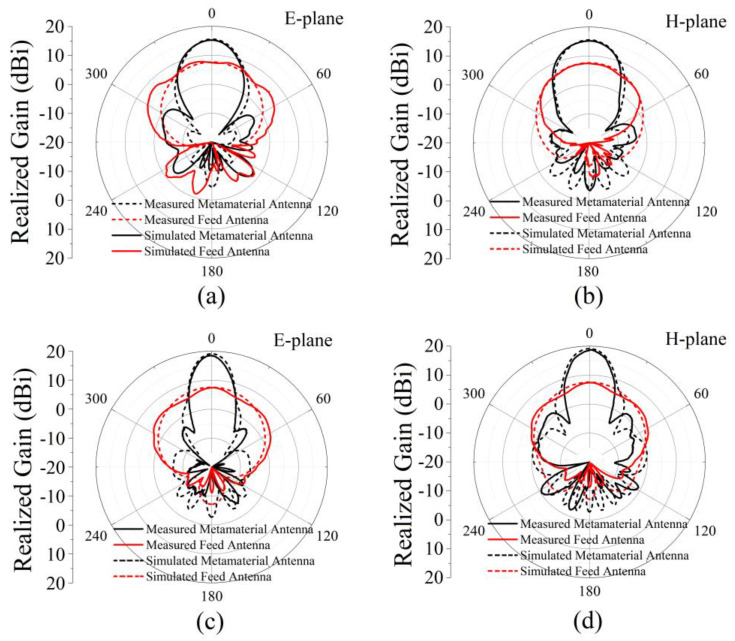
Measured (solid lines) and simulated (dashed lines) radiation patterns on the (**a**,**c**) E-plane and (**b**,**d**) H-plane. Measured results (**a**,**b**) at 3.75 GHz and (**c**,**d**) 5.60 GHz. Simulated results (**a**,**b**) at 3.73 GHz and (**c**,**d**) 5.66 GHz. The black lines are the results for the metamaterial antenna, and the red lines are the results for the feed patch antenna.

**Figure 15 micromachines-13-00058-f015:**
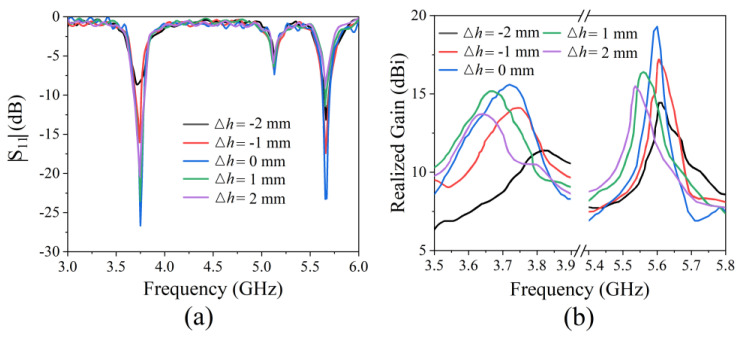
The measured radiation performance of antenna with different cavity thickness: (**a**)|S_11_|; (**b**) realized gain.

**Table 1 micromachines-13-00058-t001:** Critical dimensions of the designed dual-band AMC.

*a* (mm)	*b* (mm)	*c* (mm)	*d*_1_ (mm)	*d*_2_ (mm)	*w*_1_ (mm)	*w*_2_ (mm)	*l* (mm)
18	18	13	1	2	20	20	20

**Table 2 micromachines-13-00058-t002:** Critical dimensions of the designed feed patch.

*W_p_* (mm)	*L_p_* (mm)	*L_s_* (mm)	*W_s_* (mm)	*S_s_* (mm)	*S_f_* (mm)
19.6	22.9	18.9	0.5	0.3	9.1

**Table 3 micromachines-13-00058-t003:** Dual-band antennas performance comparison.

	This Study	REF [38]	REF [39]	REF [40]	REF [41]	REF [42]
Type of the antenna	Metamaterial antenna	SIW antenna	SIW antenna	SIW antenna	HMSIW antenna	SIW antenna
Frequency (GHz)	(1) 3.75 (2) 5.66	(1) 0.915 (2) 1.470	(1) 8.63–8.88 (2) 9.26–9.66	(1) 9.5 (2) 10.5	(1) 5.2 (2) 5.8	(1) 8.50–8.70 (2)13.03–13.84
Max Gain (dBi)	(1) 15.2 (2) 18.9	(1) −25.2 (2) −19.8	(1) 3.51 (2) 4.10	(1) 5.75 (2) 5.95	(1) 3.31 (2) 4.16	(1) 5.1 (2) 6.3

## Data Availability

The data that support the plots within this paper and other findings of this study are available from the corresponding authors on reasonable request.

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
