# Peer review of "A Dual-Band High-Gain Subwavelength Cavity Antenna with Artificial Magnetic Conductor Metamaterial Microstructures"

_micromachines, 2021, doi:10.3390/mi13010058_

Round 1

Reviewer 1 Report

This is an interesting paper. The authors presented an interesting study on implementing dual band radiating element with dual resonances metamaterial. I have certain points that can enhance it is clarity.

  • In line 107, it was mentioned that the unit cell is composed of five layers. However, the number of layers is usually counted based on the number of metallic layers. Based on Figure 2, the number of layers is 3. I suggest applying that correction to Figure 2b, and in text (from line 106 to 112).
  • In line 115, the authors mentioned that for port1 “the reference plane is added at the bottom of the circular metal patch…and port2 of the waveguide is positioned at a distance away from the top of the square metal patch with a circular slot”. Is it possible to revisit Figure 2a, and see if you can show the reference plane for both ports in the figure? Also, why your reference plane for port 2 is at a distance away from the metallic slot? Should not be referenced at the metallic slot?
  • Figure 3, 5, 7, 8, 9 , 11 and 12 are too large, and they are exceeding the text margin. It would look better if they can be resized.
  • In subsection 3.1, I suggest removing the word “theoretical” of the subsection title, and remove similar words in line 181, and 210.  This said because there is no theory demonstrated in that section, it was only simulated results that related to the Fabry-Perot cavity, which already demonstrated in section 2.
  • Mistyping in line 203 and 241. The frequency should be 5.6 GHz instead of 3.73 GHz.
  • Could you please add the simulated results (in dotted or dashed line) in Figure 11?

Reviewer 2 Report

The authors introduced an artificial magnetic conductor metamaterial for in-phase reflection and reduced the thickness of cavity antenna from lambda/2 to lambda/4.  How does this approach compare to filling the cavity with high dielectric constant material? The latter could give you even lower profile, and works for a broad bandwidth. What design considerations need to be made to increase the operating bandwidth?

Reviewer 3 Report

Authors have proposed dual-band high-gain subwavelength cavity antenna was developed and studied. they obtained AMC metamaterials that can be equivalent to MNMs at two frequencies by using the principle of multiple resonance coupling. A subwavelength cavity antenna is constructed by using this dual-band AMC metamaterial plate and a dual-band feed patch antenna. Moreover, they validated the concept with experimental results.

  1. Literature portion: This section seems bit week, since authors did not include the recent advancement of the cavity backed antenna structure. Specially authors need to incorporate the substrate Integrated waveguide-based design. Iterate the advantage and disadvantage clearly. Include the few recent advancements based SIW:

1.) Bozzi M, Georgiadis A, Wu K. Review of substrate-integrated waveguide circuits and antennas. IET Microwaves, Antennas & Propagation. 2011 Jun 6;5(8):909-20.

2.) Wideband HMSIW-based slotted antenna for wireless fidelity application." IET Microwaves, Antennas & Propagation 13, no. 2 (2019): 258-262.

3.) A coplanar‐waveguide‐fed planar integrated cavity backed slotted antenna array using TE33 mode." International Journal of RF and Microwave Computer‐Aided Engineering 30, no. 10 (2020): e22344.

4.) Wideband circular cavity‐backed slot antenna with conical radiation patterns." Microwave and Optical Technology Letters 62, no. 6 (2020): 2390-2397.

  1. The dimensions of the designs are extremely large as compared to operating frequency. Kindly validate the same.

  1. Need proof read for grammatical mistakes, for ex. In line 33 spelling is wrongly typed. Many typo mistake throughout the manuscript has been observed.

  1. Radiation patterns in both operating bands are quite different, why??? Not mentioned anywhere. It quite narrow band, application supposed to be discussed.Higher band shows a additional resonance, while its not available in lower band, why?

  1. In order to justify the performance of the proposed design, a Performance Comparison table need to be included. Kindly refer recent advancement of dual band antenna. Compare the electrical parameters like electrical size in terms of lambada, GAIN and BW of recent dual band antennas.

 (1)  A. Iqbal, M. Al-Hasan, A. Basir, B. Mabrouk, M. Nedil, and H. Yoo, "Biotelemetry and Wireless Powering of Biomedical Implants Using a Rectifier Integrated Self-Diplexing Implantable Antenna," in IEEE Transactions on Microwave Theory and Techniques, vol. 69, no. 7, pp. 3438-3451, July 2021.

(2) A. Kumar and A. A. Althuwayb, “SIW Resonator Based Duplex Filtenna,” IEEE Antennas Wirel. Propag. Lett., pp. 1–1, 2021. DOI 10.1109/LAWP.2021.3118566.

(3) Khan, A.A. and Mandal, M.K. "Compact self-diplexing antenna using dual-mode SIW square cavity." IEEE Antennas and Wireless Propagation Letters 18, no. 2 (2019): 343-347.

(4) Chaturvedi, D. and Raghavan, S. "Design and experimental verification of dual-fed, self-diplexed cavity-backed slot antenna using HMSIW technique." IET Microwaves, Antennas & Propagation 13, no. 3 (2019): 380-385.

  1. It seems that the proposed antenna has a bit complex design, as compared to previously validated dual band designs like: Dual-frequency SIW-based cavity-backed antenna. AEU [International Journal of Electronics and Communications], 97, 195–201. doi.org/10.1016/j.aeue.2018.10.019

Therefore, need to address the novelty and contribution of the proposed design clearly.

  1. Ultimately, the operation speed of the system must be quantitatively and qualitatively evaluated by considering the other analogous technologies that have been proposed for this purpose.

  1. What are other feasible alternatives? What are the advantages of adopting this technique over others in this case? How will this affect the results? The authors should provide more details on this.

  1. Some assumptions are stated in various sections. Justifications should be provided on these assumptions. Evaluation on how they will affect the results should be made.

  1. Authors carried out the simulation by using (CST Microwave studio), define the process to define the unit cell
  2. How the miss alignment of the layer affects the radiation performance, kindly include that study.
  3. Include the effect of airgap as well.
  4. Include the comparative study in the radiation graph. Include simulated and measured results together.

Reviewer 4 Report

In line #136, perhaps replace the word "minimal" with "relatively large and negative" for clarity. I can find no other areas to improve. This technical paper is very well written. 

Round 2

Reviewer 2 Report

The authors have addressed my concerns. I recommend the manuscript to be published in Micromachines. 

Reviewer 3 Report

Authors have addressed all the comments carefully.

Manuscript is

suitable for publication in current for.